# SWIFT: Mapping Sub-series with Wavelet Decomposition Improves Time Series Forecasting

## Abstract

In this paper, we propose *SWIFT*, a lightweight model that is not only powerful, but also efficient in deployment and inference for Long-term Time Series Forecasting (LTSF). Our model is based on two key points: 1. decomposition of sequences using wavelet transform. 2. using only one shared single layer for sub-series' mapping. We conduct comprehensive experiments, and the results show that *SWIFT* achieves state-of-the-art (SOTA) performance on multiple datasets, offering a promising method for edge computing and deployment in this task. Moreover, it is noteworthy that the number of parameters in *SWIFT* is only 25% of what it would be with a single-layer linear model for time-domain prediction.

## 1 Introduction

Long-term time series forecasting (LTSF) finds broad applications across various domains, including energy management, financial market analysis, weather prediction, traffic flow monitoring, and healthcare monitoring. Accurate prediction is crucial for them. At the same time, many applications require real-time prediction on edge devices, e.g., in latency-sensitive tasks such as energy scheduling or intelligent transportation systems, where models need to be responsive to real-world demands, and where edge computing and fast inference are critical. Additional challenges are posed under conditions of limited computational resources (Deng et al., 2024).

Achieving precise forecasts typically relies on powerful yet complex deep learning models, such as RNNs (Grossberg, 2013), LSTMs (Hochreiter & Schmidhuber, 1997), TCNs (Hewage et al., 2020; Wu et al., 2022), and Transformers (Zhou et al., 2021; Wu et al., 2021; Zhou et al., 2022). Thanks to the self-attention mechanism, Transformers can capture long-range dependencies in sequences, which improves prediction accuracy and makes it the most powerful of the existing LTSF forecasters. However, these models also encounter several challenges stemming from their computational complexity and the large scale of their model weights, which restrict their practical applicability, particularly in environments with limited computational resources.

Since recently a solid paper (Zeng et al., 2023) shows that even a simple one-layer linear model can outperforms Transformer-based models in almost all cases, more and more efficient linear forecasters are proposed (Das et al., 2023; Liu et al., 2023; Xu et al., 2024). While improving prediction accuracy, these linear forecasters are constantly becoming more efficient, with faster inference speed and less deployment costs, pushing the boundary of this field forward. Recently, FITS (Xu et al., 2024) modeling time-series with a complex-valued neural network, surpassed several existing Transformer models in both inference speed and forecasting performance with $10k$ parameters, establishing itself as a benchmark in the field.

However, one disadvantage of interpolation models like FITS is that the output will contain the length of the input, which creates parameter redundancy for cases where the input window is long. Besides, most current lightweight models are based on Fast Fourier Transforms (FFT) which can be used to extract periodic features and trends in time-series. But FFT is not suitable for handling non-smooth data, and the data generated in real-world applications are generally non-stationary.

Motivated by the above observations, we present *SWIFT*, a lightweight model based on first order wavelet transform Gupta et al. (2021) and only one linear layer. For the first time, we deal with the time-series only in the time-frequency domain, replacing the FFT with Discrete Wavelet Transform (DWT). *SWIFT* achieves good performance on both smooth and non-smooth data, and it is approx-

imately 100K times lighter than some mainstream models. Even when compared to a lightweight model like FITS, our model still has only 15% of its number of parameters.

In summary, our contributions can be delineated as follows:

- We propose *SWIFT*, a powerful lightweight model for time-series forecasting tasks, which is four times smaller than the single-layer linear model for time-domain prediction.
- We present the first method for processing time-series only in the time-frequency domain, applicable to both smooth and non-smooth series.
- We show that wavelet coefficients in different frequency bands can be mapped through the same representation space after aggregating features in a convolution layer.
- We conduct extensive experiments on predicting long multivariate sequences on several real-world benchmarks showing the superiority of our method. In addition, we have done experiments on anomaly detection tasks, demonstrating the strong generalization of our method.

## 2 RELATED WORK

### 2.1 EFFICIENT LINEAR FORECASTERS

Since (Zeng et al., 2023) shows that a simple one-layer linear model can outperforms Transformer forecasters (Zhou et al., 2021; Wu et al., 2021; Zhou et al., 2022) in almost all cases, there has been a rapid emergence of linear forecasters (Oreshkin et al., 2020; Das et al., 2023; Liu et al., 2023) in LTSF. The impressive performance and efficiency continuously challenge this direction.

Recently, FITS (Xu et al., 2024) introduced a frequency-domain interpolation strategy that utilizes low-pass filters and FFT (Brigham & Morrow, 1967) for time series modeling. By maintaining a parameter scale at $10k$ level, FITS surpassed several existing Transformer models in both inference speed and forecasting performance, establishing itself as a benchmark in the field.

However, FFT assumes that signals are stationary, limiting its ability to capture the temporal localization of transient or non-stationary signals (Liu et al., 2022). Furthermore, its approach of applying a global frequency domain transformation to the entire signal often results in suboptimal performance when dealing with strong boundary effects. Meanwhile, the limited representational capacity of a single-layer linear model typically necessitates a longer look-back window to prevent underfitting and distribution shifts. The parameter count of FITS is primarily determined by the length of the look-back window due to its interpolation-based prediction approach. Consequently, The efficiency of FITS decreases significantly as the lookback window increases.

Our proposed model, SWIFT, aims to enhance the field of efficient time series forecasting through the introduction of DWT. This approach not only improves SWIFT's capacity to handle non-stationary signals but also significantly reduces model's parameter count, thereby enhancing efficiency while preserving predictive performance.

### 2.2 DWT METHOD

As a powerful method for time-frequency analysis, DWT is widely used in tasks dealing with time series. (Yang et al., 2022) decomposed the time-series using wavelet decomposition and then utilized CNN and LSTM for prediction. FEDformer (Zhou et al., 2022) combines Wavelet Transform with frequency enhanced strategy and attention mechanism to capture long range dependencies. Sasal et al. (2022) utilize a maximal overlap discrete wavelet transformation and build a local transformer model for time-series forecasting. Besides, DWT is often used for anomaly detection. For instance, (Bhattacharya et al., 2022) used wavelet transform for signal denoising and damage localization. Recently, (Arabi et al., 2024) proposed a method for data augmentation in time-series prediction tasks, which is used to obtain more diverse sequences by eliminating or swapping wavelet coefficients.

However, none of these approaches provided significant insight into the wavelet transform for LTSF. In our work, we explored the wavelet coefficients in depth and found that the high-frequency coefficients and low-frequency coefficients of the historical wavelet can be mapped to the coefficients of the future wavelet in the same representation space.

## 3 PRELIMINARY

### 3.1 LTSF PROBLEM DEFINITION

In multivariate LTSF, time series data contain multiple variables or channels at each time step. Given historical values $\mathcal{X} = \{\mathbf{x}_1, \ldots, \mathbf{x}_{L_x} \mid \mathbf{x}_i \in \mathbb{R}^d\}$ where $d$ represents the number of variables and $L_x$ represents the length of the lookback window, the goal of LTSF is to predict future values $\mathcal{Y} = \{\mathbf{y}_1, \ldots, \mathbf{y}_{L_y} \mid \mathbf{y}_i \in \mathbb{R}^d\}$. The output length $L_y$ is usually much longer than the length of the lookback window $L_x$, and the feature dimension is not limited to univariate case ($d \geq 1$).

### 3.2 DWT AND TIME-FREQUENCY DOMAIN

With the discrete wavelet transform, the signal is decomposed into a series of linear combinations of wavelet functions and scale functions, the coefficients of each combination being the wavelet coefficients. In DWT, the scale function $\phi(t)$ and wavelet function $\psi(t)$ are related as follows:

$$\phi(t) = \sum_n h_\phi[n]\sqrt{2}\phi(2t - n)$$
$$\psi(t) = \sum_n h_\psi[n]\sqrt{2}\phi(2t - n) \tag{1}$$

At the same time, we are able to obtain recursive formulas for approximation coefficients $W_\phi[j, k]$ and detail coefficients $W_\psi[j, k]$, where j denotes the order of the wavelet decomposition and k denotes the shift in the time domain.

$$W_\phi[j, k] = h_\phi[-n] * W_\phi[j + 1, n]$$
$$W_\psi[j, k] = h_\psi[-n] * W_\phi[j + 1, n] \tag{2}$$

In SWIFT, We choose the Haar wavelet and perform only the first order decomposition (j = 1), and its filters corresponding to the scale and wavelet functions are:

$$h_\phi[n] = \{1/\sqrt{2}, 1/\sqrt{2}\}$$
$$h_\psi[n] = \{1/\sqrt{2}, -1/\sqrt{2}\} \tag{3}$$

We select Haar because it can make the transform fast and stable, which increases the speed of reasoning across our framework.

## 4 PROPOSED METHOD

### 4.1 STRUCTURE OVERVIEW

We propose the *SWIFT* which is shown in Figure 1. Firstly, the time-series is decomposed by $1_{st}$ order DWT. Then the high-frequency component and the low-frequency component are concatenated and mapped using the same linear layer after passing through the convolutional layer. Finally, the prediction is obtained by performing IDWT on the new components obtained from the mapping.

### 4.2 SWIFT COMPONENTS

**DWT decomposition**  Given an input sequence $\mathbf{X} \in \mathbb{R}^{N \times T}$, where $T$ is the length of lookback window, and $N$ is the number of variables, we apply a single-level DWT:

$$\mathcal{Y}_\mathcal{L}, \mathcal{Y}_\mathcal{H} = \text{DWT}(\mathbf{X}) \tag{4}$$

where $\mathcal{Y}_\mathcal{L} \in \mathbb{R}^{N \times T/2}$ represents the approximation coefficients (low-frequency components), and $\mathcal{Y}_\mathcal{H} \in \mathbb{R}^{N \times T/2}$ represents the detail coefficients (high-frequency components). The low-frequency component, obtained by convolving the input signal with a low-pass filter, captures the overall trend and smooth variations in the original signal. The high-frequency component, obtained by convolving

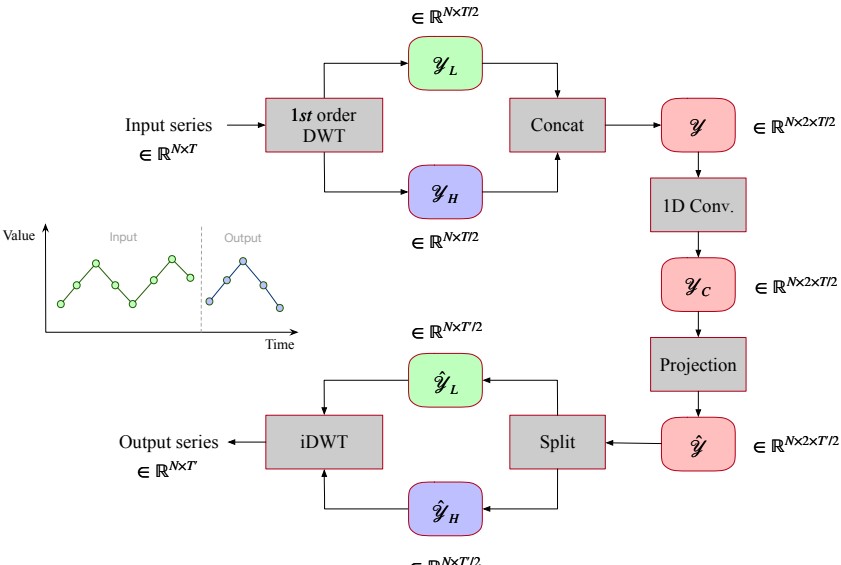

Figure 1: Overall structure of *SWIFT*. All mapping and transformation operations are represented by gray boxes. The low-frequency component, the high-frequency component, and the mapped transition component are represented by green, blue, and pink boxes, respectively.

the input signal with a high-pass filter, represents the rapid variations, discontinuities, and fine-scale structures in the signal.

We employ a novel sub-series mapping strategy that leverages the multi-resolution analysis capabilities of DWT. This approach allows us to capture and project both low-frequency trends and high-frequency details of the input time series efficiently. Our key innovation lies in the unified mapping of both low and high-frequency components. We concatenate these two components along a new dimension to gain the time-frequency representation of whole series:

$$\mathbf{Y} = [\mathcal{Y}_{\mathcal{L}}; \mathcal{Y}_{\mathcal{H}}] \in \mathbb{R}^{N \times 2 \times T/2} \tag{5}$$

After obtaining the representation $\mathbf{Y}$, SWIFT extracts information from this representation by means of convolution and Mapping. We will elaborate on this in section 4.2 and section 4.2.

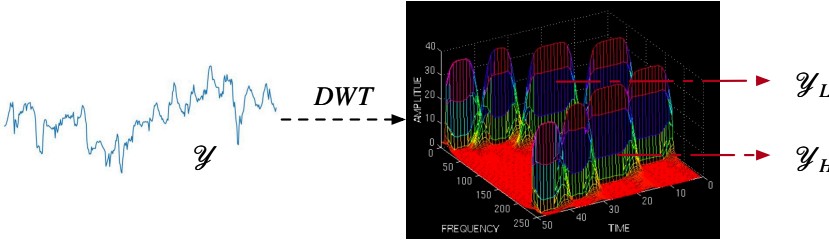

Figure 2: We use discrete wavelet transform to divide original series into low-frequency component and high-frequency component.

**Convolution layer**   In our experiments, it has been found that there is some commonality in the different band coefficients, with the potential to map through the same representation space. In addition, the timing characteristics of the coefficient vector need to further aggregated. Therefore, we add a 1D convolutional layer with input channel 2 and output channel 2. By presetting the kernel size and stride length, the sequence length before and after convolution remains constant.

In *SWIFT*, the convolution layer has three main functions: (i) Convolution layer can denoise the signal and enables feature enhancement. (ii) Convolution layer can aggregate local information and capture long and short term dependencies of time-series. (iii) Convolutional layers enable cross-band information fusion, facilitating coefficients from different bands to share a linear layer.

After the aggregation of information, we are able to get new components:

$$\mathbf{Y_C} = Conv(\mathbf{Y}) + \mathbf{Y} \in \mathbb{R}^{N \times 2 \times T/2} \tag{6}$$

**Sub-series mapping strategy**    As mentioned in the previous sections, we use DWT to handle the time series and divide it into two sub-series ($\mathcal{Y}_\mathcal{L}$ and $\mathcal{Y}_\mathcal{H}$) by $1st$ order decomposition. The obtained components are concatenated as the time-frequency representation of whole sequence $\mathbf{Y}$.

Single-layer linear models, despite their widespread use, are constrained by inherent limitations in their representational capacity. These limitations often manifest as underfitting or overfitting phenomena, particularly when applied to complex, non-stationary time series data. Such models are susceptible to being disproportionately influenced by specific patterns within the data, potentially leading to degraded predictive performance.

To address these challenges and enhance the robustness of single-layer linear models, while simultaneously reducing the model's parameter count and improving inference speed, we propose a novel mapping strategy. Our approach employs a shared weight matrix for mapping both low-frequency and high-frequency components of the input series:

$$\mathbf{Y}' = \mathbf{Y_C}\mathbf{W} + \mathbf{b} \tag{7}$$

where $\mathbf{W} \in \mathbb{R}^{T/2 \times T'/2}$ is the weight matrix, and $\mathbf{b} \in \mathbb{R}^{T'/2}$ is the bias vector. The resulting $\mathbf{Y}' \in \mathbb{R}^{N \times 2 \times T'/2}$, where $T'$ is the prediction length. After mapping, we reshape $\mathbf{Y}'$ back into approximation and detail coefficients, and apply the Inverse Discrete Wavelet Transform (IDWT) to obtain the final prediction:

$$\mathbf{Y'_L} = \mathbf{Y}'_{:,0,:}, \mathbf{Y'_H} = \mathbf{Y}'_{:,1,:}$$
$$\hat{\mathbf{Y}} = \text{IDWT}(\mathbf{Y'_L}, \mathbf{Y'_H}), \hat{\mathbf{Y}} \in \mathbb{R}^{N \times T'} \tag{8}$$

The shared mapping strategy has several advantages. (1) It enhances the robustness of the model by jointly handling the low and high frequency components, reduces the sensitivity to the presence of a single specific pattern in the time series, and mitigates the occurrence of overfitting. The shared weights encourage the model to learn generalized features applicable to both frequency ranges, thus enhancing the robustness of the model. (2) It improves parameter efficiency by using a single weight matrix for both components, which significantly reduces the total number of parameters, improves the computational efficiency of the model and speeds up inference time compared to mapping the low-frequency and high-frequency components separately. Together, these advantages enhance the prediction performance in time series forecasting involving complex, non-stationary data.

## 5 EXPERIMENT

### 5.1 FORECASTING RESULTS

Our proposed model framework aims to improve performance and efficiency in LTSF, and we thoroughly evaluate SWIFT on various time series forecasting applications.

**Datasets**    We extensively include 7 real-world datasets in our experiments, including, Traffic, Electricity, Weather, ETT (4 subsets) used by Autoformer (Wu et al., 2021). We summarize the characteristics of these datasets in appendix.

**Baselines**    We carefully choose well-acknowledged forecasting models as our benchmark, including (1) Transformer-based methods: FEDformer (Zhou et al., 2022) and PatchTST (Nie et al., 2023).

(2) Efficient Linear-based methods: DLinear (Zeng et al., 2023), FITS (Xu et al., 2024). (3): TCN-based methods: TimesNet (Wu et al., 2022). We rerun all the experiment with code and script provided by their official implementation.

**Implementation details** Our method is trained with the ADAM optimizer (Kingma & Ba, 2015), using OneCycleLR strategy to adjust the learning rate. We fix the length of lookback window as 720, and we set the predicted window among {96, 192, 336, 720}. As for the kernel size of the convolution layer, we choose suitable size in {3, 9, 13, 17}.

**Evaluation** To avoid information leakage, We choose the hyper-parameter based on the performance of the validation set. We follow the previous works (Zhou et al., 2021; Zeng et al., 2023; Xu et al., 2024) to compare forecasting performance using Mean Squared Error (MSE) as the core metrics.

**Main results** Comprehensive forecasting results are listed in Table 1 and Table 2 with the best in **bold** and the second underlined. The lower MSE indicates the more accurate prediction result. As shown in table 1 and table 2, SWIFT performs well in the forecasting task. Compared to FITS, the most powerful and efficient model currently, SWIFT achieves comparable or even superior performance in all 7 datasets.

Table 1: Long-term forecasting results with fixed lookback length $T = 720$ on ETT dataset in MSE. The best result is highlighted in **bold**, and the second best is highlighted with underline. IMP is the improvement between SWIFT and the second best/ best result, where a larger value indicates a better improvement. Most of the STD are under 5e-4 and shown as 0.000 in this table.

| Dataset | ETTh1 | | | | ETTh2 | | | | ETTm1 | | | | ETTm2 | | | |
|---|---|---|---|---|---|---|---|---|---|---|---|---|---|---|---|---|
| Horizon | 96 | 192 | 336 | 720 | 96 | 192 | 336 | 720 | 96 | 192 | 336 | 720 | 96 | 192 | 336 | 720 |
| FEDFormer | 0.375 | 0.427 | 0.459 | 0.484 | 0.340 | 0.433 | 0.508 | 0.480 | 0.362 | 0.393 | 0.442 | 0.483 | 0.189 | 0.256 | 0.326 | 0.437 |
| TimesNet | 0.384 | 0.436 | 0.491 | 0.521 | 0.340 | 0.402 | 0.452 | 0.462 | 0.338 | 0.374 | 0.410 | 0.478 | 0.187 | 0.249 | 0.321 | 0.408 |
| Dlinear | 0.384 | 0.443 | 0.446 | 0.504 | 0.282 | 0.350 | 0.414 | 0.588 | 0.301 | 0.335 | 0.371 | 0.426 | 0.171 | 0.237 | 0.294 | 0.426 |
| PatchTST | 0.385 | 0.413 | 0.440 | 0.456 | 0.274 | 0.338 | 0.367 | 0.391 | **0.292** | **0.330** | 0.365 | 0.419 | 0.163 | 0.219 | 0.276 | 0.368 |
| FITS | 0.379 | 0.413 | 0.433 | **0.430** | 0.271 | 0.331 | 0.354 | **0.377** | 0.302 | 0.337 | 0.366 | 0.415 | 0.162 | 0.216 | 0.268 | **0.348** |
| SWIFT | **0.367** | **0.402** | **0.429** | 0.433 | **0.267** | **0.328** | **0.351** | 0.381 | 0.305 | 0.335 | **0.363** | **0.411** | **0.161** | **0.214** | **0.267** | **0.348** |
| STD | 0.000 | 0.000 | 0.000 | 0.000 | 0.000 | 0.000 | 0.000 | 0.000 | 0.000 | 0.000 | 0.000 | 0.000 | 0.000 | 0.000 | 0.000 | 0.000 |
| IMP | 0.002 | 0.011 | 0.004 | -0.003 | 0.004 | 0.003 | 0.03 | -0.004 | -0.013 | -0.005 | 0.003 | 0.004 | 0.001 | 0.002 | 0.001 | 0.000 |

Table 2: Long-term forecasting results with fixed lookback length $T = 720$ on three popular datasets in MSE. The best result is highlighted in **bold** and the second best is highlighted with underline. IMP is the improvement between SWIFT and the second best/ best result, where a larger value indicates a better improvement. Most of the STD are under 5e-4 and shown as 0.000 in this table.

| Dataset | Weather | | | | Electricity | | | | Traffic | | | |
|---|---|---|---|---|---|---|---|---|---|---|---|---|
| Horizon | 96 | 192 | 336 | 720 | 96 | 192 | 336 | 720 | 96 | 192 | 336 | 720 |
| FEDformer | 0.246 | 0.292 | 0.378 | 0.447 | 0.188 | 0.197 | 0.212 | 0.244 | 0.573 | 0.611 | 0.621 | 0.630 |
| TimesNet | 0.172 | 0.219 | 0.280 | 0.365 | 0.168 | 0.184 | 0.198 | 0.220 | 0.593 | 0.617 | 0.629 | 0.640 |
| Dlinear | 0.174 | 0.217 | 0.262 | 0.332 | 0.140 | 0.153 | 0.169 | 0.204 | 0.413 | 0.423 | 0.437 | 0.466 |
| PatchTST | 0.151 | 0.195 | 0.249 | 0.321 | **0.129** | 0.149 | 0.166 | 0.210 | **0.366** | **0.388** | **0.398** | 0.457 |
| FITS | 0.143 | 0.186 | **0.236** | **0.307** | 0.134 | 0.149 | 0.165 | **0.203** | 0.385 | 0.397 | 0.410 | **0.448** |
| SWIFT | **0.142** | **0.185** | **0.236** | 0.308 | 0.133 | **0.148** | **0.164** | **0.203** | 0.385 | 0.396 | 0.410 | **0.448** |
| STD | 0.000 | 0.000 | 0.000 | 0.000 | 0.000 | 0.000 | 0.000 | 0.000 | 0.000 | 0.000 | 0.000 | 0.000 |
| IMP | 0.001 | 0.001 | 0.000 | -0.001 | -0.004 | 0.001 | 0.001 | 0.000 | -0.019 | -0.008 | -0.012 | 0.009 |

Table 3 presents the number of trainable parameters and MACs for various Linear-based time series forecasting (TSF) models using a look-back window of 720 and a forecasting horizon of 96 on the Electricity dataset. The table clearly demonstrates the exceptional efficiency of SWIFT compared to other models. Among all efficient models, SWIFT stands out with significantly fewer parameters and much faster training times. SWIFT requires only 15% of the parameters of FITS and 60% of its MACs, while achieving comparable or even superior performance to these state-of-the-art efficient forecasting models. It is worth emphasizing that SWIFT's parameter count is also much lower than Dlinear, which has 139.7K parameters. Moreover, while FITS' parameter count increases rapidly when using long look-back windows for forecasting, SWIFT does not exhibit this issue.

Table 3: Number of trainable parameters, MACs, and training time of Linear-based models under look-back window=720 and forecasting horizon=96 on the Electricity dataset.

| Model | Parameters | MACs | Train./epoch (GPU) | Train./epoch (CPU) |
|-------|-----------|------|--------------------|--------------------|
| DLinear | 138.4k | 1449.58 M | *15.811s* | *27.307s* |
| FITS | 115.8k | 4759.63 M | *25.070s* | *75.339s* |
| SWIFT | 17.4k | 2854.79 M | *16.833s* | *65.853s* |

**Ablations**  We did two main ablation experiments, which is shown in table 4 and table 5.

We chose four datasets from ETT for the convolution layer ablation experiments. Obviously, the role of the convolution layer in our model is crucial, which can be proved by the overall increase in performance. In SWIFT, the convolution layer not only denoises sequences and captures timing dependencies, but also enables cross-band feature integration.

Table 4: Model performance before and after adding convolution layers. IMP is the improvement between with $Conv.$ and without $Conv.$, where a larger value indicates a better improvement.

| Dataset | ETTh1 | | | | ETTh2 | | | | ETTm1 | | | | ETTm2 | | | |
|---------|------|------|------|------|------|------|------|------|------|------|------|------|------|------|------|------|
| Horizon | 96 | 192 | 336 | 720 | 96 | 192 | 336 | 720 | 96 | 192 | 336 | 720 | 96 | 192 | 336 | 720 |
| +$Conv.$ | **0.367** | **0.402** | **0.429** | **0.433** | **0.267** | **0.328** | **0.351** | **0.381** | **0.305** | **0.335** | **0.363** | **0.411** | **0.161** | **0.214** | **0.267** | **0.348** |
| $w/o$ | 0.372 | 0.418 | 0.446 | 0.446 | 0.276 | 0.337 | 0.360 | 0.388 | 0.310 | 0.346 | 0.367 | 0.414 | 0.164 | 0.216 | 0.273 | 0.352 |
| IMP | 0.005 | 0.016 | 0.017 | 0.013 | 0.009 | 0.009 | 0.009 | 0.007 | 0.005 | 0.011 | 0.004 | 0.003 | 0.003 | 0.002 | 0.006 | 0.004 |

Table 5 shows that the high-frequency and low-frequency components obtained after wavelet decomposition are able to share a linear layer for mapping, which does not result in performance loss. After conducting an in-depth study, we came to the following two conclusions: (i) The components of different frequency bands obtained after DWT may have some underlying feature correlation, so they can be represented and mapped in the same feature space. (ii) Convolution layer enables cross-band feature fusion, which is shown in table 4.

Table 5:  IMP is the improvement between with $Share$ and $Split$ result, where a larger value indicates a better improvement. $Share$ corresponds to the use of only one linear layer, while $Split$ corresponds to the use of one linear layer for each frequency band.

| Dataset | ETTh1 | | | | ETTh2 | | | | ETTm1 | | | | ETTm2 | | | |
|---------|------|------|------|------|------|------|------|------|------|------|------|------|------|------|------|------|
| Horizon | 96 | 192 | 336 | 720 | 96 | 192 | 336 | 720 | 96 | 192 | 336 | 720 | 96 | 192 | 336 | 720 |
| $Share$ | **0.367** | **0.402** | 0.429 | **0.433** | 0.267 | **0.328** | **0.351** | 0.381 | **0.305** | 0.335 | **0.363** | **0.411** | **0.161** | 0.214 | 0.267 | **0.348** |
| $Split$ | 0.368 | **0.402** | **0.428** | 0.434 | **0.266** | 0.329 | 0.353 | **0.380** | **0.305** | **0.334** | **0.363** | 0.412 | 0.162 | **0.212** | **0.266** | **0.348** |
| IMP | 0.001 | 0.000 | -0.001 | 0.001 | -0.001 | 0.001 | 0.002 | -0.001 | 0.000 | -0.001 | 0.000 | 0.001 | 0.001 | -0.002 | -0.001 | 0.000 |

## 5.2 ANOMALY DETECTION RESULTS

**Datasets**  We use 4 benchmark datasets used by (Xu et al., 2022): SMD (Server Machine Dataset), PSM (Polled Server Metrics), MSL (Mars Science Laboratory rover) and SMAP (Soil Moisture Active Passive satellite).

**Baseline**  We compare SWIFT with models such as TimesNet (Wu et al., 2022), Anomaly Transformer (Xu et al., 2022), THOC (Shen et al., 2020), Omnianomaly (Su et al., 2019), DGHL (Challu et al., 2022). Following TimesNet, we also compare the anomaly detection performance with other models (Zeng et al., 2023).

**Main results**  Table 6 demonstrates the performance of SWIFT on various datasets.SWIFT achieves near-perfect F1 scores of 99.92% and 96.487% on the SMD and PSM datasets, respectively, and achieves about 2.5% outperformance on PSM compared to FITS, demonstrating its high-precision performance in anomaly detection. By introducing the time-frequency domain, SWIFT can easily identify the anomalies that are difficult to recognize in the time domain, including the

anomalies that introduce unexpected time-frequency components. In contrast, models such as TimesNet, Anomaly Transformer and Stationary Transformer do not perform as well as SWIFT on these datasets.

However, SWIFT performs relatively low on the SMAP and MSL datasets. Due to the binary event data nature of these datasets, time-frequency feature representations may have difficulty capturing these features effectively. In this case, time-domain modeling is more desirable because the raw data format is already compact enough. As a result, models designed for anomaly detection, such as THOC and Omni Anomaly, achieve higher F1 scores on these datasets.

It is worth noting that SWIFT has an extremely low number of parameters ranging from 0.625k-2.5k for the anomaly detection task. Such a feature allows SWIFT to be deployed on almost any edge device. In addition, SWIFT's inference speed is impressive, reaching sub-millisecond levels, much faster than the latency associated with larger models or communication overheads. This speed underscores SWIFT's suitability as a first-response tool for rapid detection of critical errors.

Table 6: Anomaly detection result of F1-scores on 4 datasets. The best result is highlighted in **bold**, and the second best is highlighted with underline.

| Models | SWIFT | FITS | TimesNet | Anomaly | THOC | Omni | Stationary | DGHL | OCSVM | IForest | LightTS | Dlinear |
|---|---|---|---|---|---|---|---|---|---|---|---|---|
| SMD | 99.92 | **99.95** | 85.81 | 92.33 | 84.99 | 85.22 | 84.72 | N/A | 56.19 | 53.64 | 82.53 | 77.1 |
| PSM | 96.47 | 93.96 | 97.47 | 97.89 | **98.54** | 80.83 | 97.29 | N/A | 70.67 | 83.48 | 97.15 | 93.55 |
| SMAP | 68.16 | 70.74 | 71.52 | **96.69** | 90.68 | 86.92 | 71.09 | 96.38 | 56.34 | 55.53 | 69.21 | 69.26 |
| MSL | 58.51 | 78.12 | 85.15 | 93.59 | 89.69 | 87.67 | 77.5 | **94.08** | 70.82 | 66.45 | 78.95 | 84.88 |

## 6 CONCLUSION AND FUTURE WORK

In this paper, we propose SWIFT for time series analysis, an efficient linear-based model with performance comparable to state-of-the-art models that are typically several orders of magnitude larger. In future work, we plan to evaluate SWIFT in more real-world scenarios, such as in anomaly detection and classification tasks, and to improve its interpretability. In addition, we plan to explore large neural networks in the time-frequency domain to perform scaling up operations on SWIFT and improve its prediction performance, such as large transformer models based on DWT.

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
