# OpenReview forum: "SWIFT: Mapping Sub-series with Wavelet Decomposition Improves Time Series Forecasting"
_ICLR.cc/2025/Conference — ICLR 2025 Conference Withdrawn Submission_

### Official Review · Reviewer_4km3 · 2024-11-01

**Soundness:** 2
**Presentation:** 1
**Contribution:** 1
**Rating:** 1
**Confidence:** 4

**Summary:**

This paper proposes a lightweight long-term time series forecasting (LTSF) model named SWIFT. The model is based on first-order discrete wavelet transform (DWT) and a single-layer linear mapping, aiming to improve prediction efficiency and performance by processing time series data in the time-frequency domain. The authors claim that SWIFT can effectively handle both smooth and non-smooth data, has significantly fewer parameters than existing mainstream models, and achieves superior prediction performance on several real-world datasets. Additionally, SWIFT is reported to perform well on anomaly detection tasks.

**Strengths:**

1) Practical Applicability: Long-term time series forecasting has widespread applications in domains such as energy management, financial market analysis, and weather prediction. Researching lightweight models that aim to improve prediction efficiency and performance has practical significance.

2) Reduced Model Size: The SWIFT model has a significantly smaller parameter count compared to some existing models, which could be advantageous in resource-constrained environments or applications requiring real-time predictions.

**Weaknesses:**

1.Lack of Novelty:Using DWT Instead of FFT is Not Novel: Employing discrete wavelet transform (DWT) as a replacement for fast Fourier transform (FFT) in time series analysis is a well-explored area. The paper does not clearly articulate its unique methodological contributions or sufficiently differentiate from existing work.

2.Insufficient Theoretical Support and Methodological Analysis:Lack of Theoretical Proof for DWT Advantages: The paper claims that DWT is better suited for handling non-smooth and non-stationary data but does not provide theoretical analysis or empirical evidence to support this claim, nor does it reference relevant literature.

3.Justification for Choosing Haar Wavelet is Missing: The use of the Haar wavelet as the sole wavelet function is not justified; there is no comparison with other wavelet functions (e.g., Daubechies, Symlet), and no experiments are provided to support this choice.

4.Unclear Role of the Convolutional Layer: While the convolutional layer is said to denoise and enhance features, there is a lack of detailed explanation of its theoretical basis or empirical evidence demonstrating its effectiveness. The supposed commonality of coefficients across different frequency bands is not clearly explained.

5.Unproven Advantages of the Shared Mapping Strategy: The proposed shared mapping strategy lacks theoretical analysis or experimental evidence to support its claimed benefits, and there are no comparative experiments with independent mapping strategies.

6.Experimental Design and Validation are Inadequate:Limited Evaluation Metrics: The experiments rely solely on Mean Squared Error (MSE) as the evaluation metric, lacking other key metrics such as Mean Absolute Error (MAE), which are essential for a comprehensive assessment of model performance.

7.Use of Outdated Baselines: The experiments do not include comparisons with the latest models from 2024, using instead older models, which diminishes the impact and relevance of the results.

8.Inconsistent Experimental Results: In some datasets and tasks, SWIFT does not significantly outperform baseline models, and sometimes performs worse. The paper lacks in-depth analysis of these results.

9.Lack of Ablation Studies and In-depth Analysis: The paper does not include ablation studies for key components of the model (e.g., DWT), failing to demonstrate the contribution of each part to overall performance.

10.Insufficient Discussion of Model Limitations:Performance on Non-stationary Data Not Analyzed: Despite claiming that the model can handle non-smooth and non-stationary data, there are no dedicated experiments or analyses to validate its effectiveness on such data.

11.Inadequate Explanation of Anomaly Detection Results: The model performs poorly on the SMAP and MSL datasets in anomaly detection tasks, but the paper does not deeply investigate the reasons or propose improvements.

12.Computational Efficiency and Parameter Count Not Convincingly Demonstrated:Efficiency Advantages are Unclear: In some cases, SWIFT's training time and computational load are not significantly better than other models, and may even be higher. The claimed efficiency gains are not sufficiently substantiated.

13.Impact of Reduced Parameter Count Not Fully Discussed: While the parameter count is reduced, the potential negative effects on the model's representational capacity and predictive performance are not explored.

**Questions:**

1.Regarding the Advantages of DWT:Can the authors provide theoretical analysis or empirical evidence demonstrating that DWT is better suited than FFT for handling non-smooth and non-stationary data? Are there relevant references supporting this claim?

2.Choice of Haar Wavelet:Why was the Haar wavelet chosen over other wavelet functions? Have experiments been conducted to compare the impact of different wavelet functions on model performance to justify this selection?

3.Role of the Convolutional Layer:What specific functions does the convolutional layer serve in the model? Can the authors provide theoretical analysis or empirical results to support its effectiveness in denoising and feature enhancement?

4.What is the theoretical basis for the supposed commonality of coefficients across different frequency bands? Can this be explained in more detail, with supporting experimental evidence?

5.Selection of Evaluation Metrics:Why was MSE chosen as the sole evaluation metric for LSTF? Have other metrics such as MAE and MAPE been considered to provide a more comprehensive evaluation of model performance?

6.Selection of Baseline Models:Have the authors considered including the latest models from 2024 for comparison to enhance the persuasiveness of the experiments?

7.Performance on Anomaly Detection Tasks:What might be the reasons for SWIFT's poor performance on the SMAP and MSL datasets? Are there any proposed solutions or further analyses?

8.Practical Application of the Model:Are there any case studies or deployments of the model in real-world scenarios or on edge devices to verify its practical value and efficiency?

---

### Official Review · Reviewer_zumf · 2024-11-03

**Soundness:** 2
**Presentation:** 2
**Contribution:** 2
**Rating:** 1
**Confidence:** 5

**Summary:**

The paper introduces SWIFT, a lightweight model for Long-term Time Series Forecasting (LTSF) that employs wavelet decomposition and a shared single layer for subseries mapping. Key contributions include a novel time-frequency domain approach applicable to both smooth and non-smooth series, a shared representation space for different frequency bands, and extensive experiments demonstrating superior performance over other models with significantly fewer parameters.

**Strengths:**

Though the use of wavelet decomposition in the context of time series forecasting is a common practice , but this work offers a fresh perspective on handling non-stationary data by employing shared convolution and linear projection. The proposed model achieving promising forecasting performance with relatively small parameter-count.

The experiments demonstrate the effectiveness of SWIFT in terms of efficiency and accuracy.

The paper is well written, with a logical flow that makes the complex concepts accessible.

The lightweight nature of SWIFT and its performance can be useful for applications requiring real-time predictions on edge devices.

**Weaknesses:**

The paper could provide more theoretical justification for the choice of wavelet decomposition and its impact on model performance.

There is room for improvement in the model's interpretability, which is crucial for understanding and trusting the model's predictions.

The proposed method does not perform well on more complex datasets, such as **ECL**, **Weather**, and **Traffic**.

The author chose to use the **ETT** dataset, which has the fewest variables, rather than more complex multivariate datasets like **Traffic** and **ECL**, to conduct ablation experiments in **Table 4&5**. This could severely damage the persuasiveness of the ablation study.

In **Table 4**, simply removing the convolutional layers may not be sufficient to constitute an appropriate ablation study, as other design choices, such as employing a linear projection, could similarly yield comparable improvements.

A crucial strategy employed in this paper is **Channel Independence**, which was proposed in **PatchTST**. However, this significant aspect is not discussed at all throughout the text.

**Questions:**

Could the authors elaborate on the theoretical underpinnings of using wavelet decomposition for time series forecasting, especially in comparison to other frequency domain methods?

---

### Official Review · Reviewer_9n4Y · 2024-11-04

**Soundness:** 2
**Presentation:** 3
**Contribution:** 2
**Rating:** 5
**Confidence:** 4

**Summary:**

The manuscript introduces a novel model named SWIFT for Long-term Time Series Forecasting (LTSF). The primary contributions are twofold: it utilizes wavelet decomposition to transfer the sequences, and then employs a 1D convolutional and another single shared layer for mapping sub-series. The results emphasize the model's efficiency and performance, claiming state-of-the-art results with reduced computational requirements compared to not only linear models, but also heavy (e.g., transformer-based) models. This innovative approach is suggested as a promising solution for applications in edge computing.

**Strengths:**

S1. This paper is a bold attempt to improve the efficiency of time series analysis tasks.

S2. The proposed method, SWIFT, is evaluated on two tasks, both forecasting and anomaly detection, across a few datasets against a collection of diverse comparison methods.

S3. The paper is in general well-written and easy-to-follow.

**Weaknesses:**

W1. Although there is a clear need for efficiency at edge computing, it is not convincing to me the choice of DWT. Breaking this point down:

W1-1. Although the discussion against FFT has been provided, it would still be very informative if the authors can provide empirical evaluation against FFT, and potentially other dimension reduction techniques, such as PCA, et al.

W1-2. It would also be useful if the author can provide experimental results to justify the choice of the Haar wavelet.

W2. Similarly to W1, it is also not straightforward to me why the design of the framework is so effective in terms of achieving comparable-to-SOTA MSE. My doubt comes from that fact that, basically, the transformation in the DWT space is just one 1D convolution and another fully-connected layer, why this simple learning can outperform many complex models, e.g., those transformer-based methods?

W3. The anomaly detection in Table 6 clearly suggests that SWIFT is more suitable in some applications while struggling on some others. This damages the versatility of SWIFT, and at the same time,

W3-1. The foresting results are only conducted on four ETTh datasets, which are from the same domain and same provider. Some other datasets, say, Weather, Traffic, Electricity, ILI in the PatchTST paper, should be included at least.

W3-2. It would be very useful for the readers to understand the pros and cons of SWIFT, why it performs so well under some scenarios, and under what specific situations should the users choose SWIFT without negative side effects.

**Questions:**

Q1. What are the general and typical requirements of efficiency for time series analysis tasks at edges? It would be very useful to evaluate the real impact of the work, if the authors can name a few widely-applicable real world scenarios, and specify what is the memory constraint, what is the inference time constraint, what is the accuracy requirement, and whether SWIFT can satisfy these requirements or if SWIFT is the only solution that can satisfy these requirements.

Q2. Some of the results reported in this paper, e.g., the PatchTST results in Table 1, are not consistent (and generally worse than) with the numbers reported in their original papers. Why this is the case?

Q3. Some minor suggestions:

Q3-1. “section 4.2” is repeated at line 198.

Q3-2. The discussion of the advantage of the convolutional layers between line 216-220 is somehow very general. Only the third point is relatively connected to the SWIFT design.

Q3-3. In line 228, why a simple single-layer linear model can be prone to overfitting?

---

### Official Review · Reviewer_mJp2 · 2024-11-08

**Soundness:** 2
**Presentation:** 3
**Contribution:** 2
**Rating:** 5
**Confidence:** 3

**Summary:**

This paper proposes a lightweight time series forecasting algorithm that transforms the data into the wavelet domain and then performs predictions through convolutions in the wavelet domain. The entire method requires fewer network parameters and achieves state-of-the-art (SOTA) experimental results. The paper is quite interesting, but the performance improvement over FITS is somewhat marginal. Additionally, the paper lacks a convincing explanation for the convolution of wavelet transform coefficients. The paper demonstrates that using some regularization methods (e.g., using convolutions instead of fully connected layers) can improve the algorithm's performance, but it lacks further explanation or analysis.

**Strengths:**

1. The proposed method in the paper is relatively simple and requires fewer parameters. It also avoids the requirement for a lookback window for supervision, unlike FITS.
2. The proposed method is inherently linear, and some regularization operations are applied through the network design. During prediction, it can be transformed into a linear function, enabling fast predictions.
3. The paper includes comprehensive and thorough experiments.

**Weaknesses:**

1. Compared to FITS, the performance improvement of the proposed method is marginal. Additionally, the reduction in the number of parameters is limited to the training process; during the prediction process, the number of parameters is comparable to that of FITS.
2. Since the final network parameter sharing can be considered as a convolution between the real and imaginary parts, the core operation of the paper is performing convolutions in the wavelet domain. However, the physical meaning of convolutions in the wavelet domain is difficult to understand, as adjacent wavelet coefficients generally have low correlation due to the orthogonality of adjacent wavelet bases. I would recommend the authors provide more in-depth analysis.

**Questions:**

1. Did the authors re-run the FITS results on the ETT dataset? The results reported by the authors are somewhat inconsistent with those in the original paper.
2. Did the authors use a 2-channel convolution? Or did they use separate channels for the high-frequency and low-frequency wavelet coefficients, resulting in two channels in total?
3. Did the authors use the RevIN normalization method? If not, how does their method address the distribution shift in time series data?
4. Is Figure 2 the result after wavelet transformation? The left side shows a one-dimensional time series, but the right side appears to be two-dimensional data. How did the authors process this two-dimensional data to convert it back to one-dimensional? Is this due to the use of a first-order transformation? In Section 2.1, the authors mention that FFT is not suitable for non-stationary data. Is wavelet transform more suitable for non-stationary data? Does a first-order wavelet transform compromise its ability to handle non-stationary data?

---

### Public Comment · ~Eamonn_Keogh1 · 2024-11-24
**You test on SMD MSL SWaT PSM. However it is increasingly understood that it is meaningless to test on these datasets**

You test on SMD MSL SWaT PSM. However it is increasingly understood that it is meaningless to test on these datasets, because the ground truth is mislabeled, they are trivial, and have other flaws.

https://www.dropbox.com/scl/fi/x6ie264xrfkl0nbdw1vtb/Irrational-Exuberance_why_most_TSAD_is_wrong.pdf?rlkey=16frcr2lo6ip5o6uf18qwdeud&dl=0
and
https://www.dropbox.com/scl/fi/cwduv5idkwx9ci328nfpy/Problems-with-Time-Series-Anomaly-Detection.pdf?rlkey=d9mnqw4tuayyjsplu0u1t7ugg&dl=0

---

> ### Author Response · Authors · 2024-11-26
> **Thanks for pointing out this issue**
>
> Thank you very much for pointing this out! We overlooked this issue previously because previous papers have used these datasets as baselines, and we hadn't found any other better datasets.

---

### Note · Authors · 2024-11-26

I have read and agree with the venue's withdrawal policy on behalf of myself and my co-authors.